

# No relationship between vertebral column shifts and limb fluctuating asymmetry in human foetuses

Clara M.A. ten Broek[1,2], Jessica Bots[2], Marianna Bugiani[3], Frietson Galis[1] and Stefan Van Dongen[2]

[1] Naturalis Biodiversity Center, Leiden, Netherlands
[2] Evolutionary Ecology Group, Department of Biology, Universiteit Antwerpen, Wilrijk, Antwerp, Belgium
[3] Department of Pathology, VU Medical Center, Amsterdam, Netherlands

## ABSTRACT

Disturbance from the normal developmental trajectory of a trait during growth—the so-called developmental instability—can be observed morphologically through phenodeviants and subtle deviations from perfect symmetry (fluctuating asymmetry). This study investigates the relationship between phenodeviance in the human vertebral column (as a result of axial patterning defects) and limb fluctuating asymmetry. Since both types of markers of developmental instability have been found associated with congenital abnormalities in humans, we anticipate a relationship between them if the concept of developmental instability, measured through either phenodeviants or asymmetry, would reflect an organism-wide process. Yet we did not find any support for this hypothesis. We argue that the vast differences in the developmental processes involved in both systems renders these two markers of developmental instability unrelated, in spite of their associations with other congenital abnormalities. Our results thus contribute to the growing awareness that developmental instability is not an organism-wide property.

## INTRODUCTION

Developmental disturbance experienced during early growth can have important consequences for the morphology, behavior, and stress tolerance of an individual later in life (e.g., *Roseboom et al., 2001*; *Fujioka et al., 2006*; *Weinstock, 2008*). One type of such developmental disturbances, developmental instability (DI), reflects the inability of an organism to undergo stable development under given environmental and genetic conditions. It is increased by (stress induced) developmental perturbations and/or the lack of efficient mechanisms stabilizing development. Levels of developmental instability (DI) can be assessed in two ways, namely the occurrence of abnormal morphological deviations and subtle differences between left and right of on average bilateral traits, which are on average symmetrical. The idea behind the use of directionally random asymmetry (i.e., fluctuating asymmetry, FA) is that the two sides of an organism represent

Corresponding authors
Clara M.A. ten Broek,
cma.ten.broek@gmail.com
Stefan Van Dongen,
Stefan.vandongen@uantwerpen.be

two replicates of the same developmental process, and any deviation from symmetry is the outcome of random noise and the inability to buffer development against it (e.g., *Ludwig, 1932*; *Van Valen, 1962*; *Palmer & Strobeck, 1986*). The occurrence or frequency of morphological abnormalities—the so-called phenodeviants, a term first coined by *Lerner (1954)*—were proposed as a measure of DI by *Rasmuson (1960)*. Phenodeviants are less often used as measure of DI and associations with FA are not always present (e.g., *Bots et al., 2016*), suggesting that both measures of DI do not necessarily reflect similar aspects of DI. Although FA is commonly used in evolutionary studies to measure DI, to date it remains an unpredictable risk marker in the sense that it does not ubiquitously relate to either environmental or genetic stress (e.g., *Møller, 1997*; *Lens et al., 2002*; *Van Dongen & Gangestad, 2011*). Since little is known about the factors that influence the strength of stress-DI associations when measured through phenodeviants or FA, it is important to gain insights in the associations between both measures of DI. In addition, when the occurrence or frequency of phenodeviants are used as indicators of DI, all morphological abnormalities are considered as being equally important or severe. This should not be true in all cases. More specifically, changes in vertebral identity along the vertebral column provide a good indicator for the length and extent of disturbed development. The different types of phenodeviance along the vertebral column indicate different levels of experienced stress or perturbations during development. The more boundaries between vertebral regions have been shifted (so called homeotic transformations), the longer the disturbance of axial patterning lasts and the more congenital abnormalities are found associated in other parts of the body (*Ten Broek et al., 2012*). Moreover, the extent of vertebral column variation indicates the vulnerability of developing mammals more generally (*Varela-Lasheras et al., 2011*; *Reumer, Ten Broek & Galis, 2014*). Thus, morphological abnormalities of the vertebral column reflect a gradient in the severity of phenodeviance, while a vertebral abnormality in itself poses no immediate functional limitations prenatally (see also below), providing an interesting model system to study its relation with FA and gain better insights in the link between the two markers of DI.

Formation of the (mesenchymal) vertebral column starts in the fourth week of development with the migration of sclerotome cells, and it is in this embryonic period (until the end of the fifth week) that the identity of the different segments of the vertebral column is being determined (e.g., *Sadler, 2011*). This determination happens as part of the early anterior-posterior patterning of the paraxial mesoderm, mediated by the well-known *Hox*-genes (e.g., *Kessel & Gruss, 1991*; *Kmita & Duboule, 2003*; *Woltering & Durston, 2008*). At this stage of development, the foetus is extremely vulnerable to environmental and genetic insults, and disruption of development may not only result in vertebral column anomalies, but also in other birth defects, because of low effective modularity (*Sander, 1983*; *Raff, 1994*; *Galis & Metz, 2001*; *Ten Broek et al., 2012*), as patterning processes of the three body axes and simultaneously occurring morphogenetic processes interact strongly (*Diez del Corral et al., 2003*; *Cordes et al., 2004*; *Aulehla & Pourquie, 2010*; *Durston et al., 2011*). The study for the association between vertebral column variation and limb FA is particularly interesting because limb formation starts by the end of the fourth week (and thus in the same embryonic period) when limb buds become visible as outpocketings

from the ventrolateral body walls. However, it is not until the sixth week of development that the hand- and footplates start to form (reviewed in *Capdevila & Belmonte, 2001*, *Moore & Persaud, 2003* and *Sadler, 2011*). Limbs then continue to grow until the stage in which we took our measurements, and thus will likely also reflect an accumulated level of developmental perturbations. More specifically, studying the association between vertebral variations and limb FA will allow us to study to what extent limb FA can be seen as a general/omnibus measure of DI, similar to the variations observed in the vertebral column."

We investigate the relationship between axial patterning abnormalities (here considered as vertebral column variation, through homeotic transformations) and limb FA in human foetuses. We use a hospital collection of human deceased foetuses of the VU University Medical Centre in Amsterdam. We determined axial patterning abnormalities in the vertebral column which can be categorised according to their development (*Ten Broek et al., 2012* and below).

While we have shown earlier that FA and vertebral abnormalities relate to—at least some—congenital abnormalities in deceased human foetuses (*Van Dongen et al., 2009*; *Bots et al., 2011*; *Ten Broek et al., 2013*), the link between FA and cervical ribs was less clear (*Van Dongen et al., 2009*).

Here we present a more detailed study on the relationship between FA and vertebral column variation focussing on the severity of phenodeviance in a much larger sample. If both limb FA and vertebral column variation reflect individual DI, we predict a higher limb FA in foetuses with vertebral column abnormalities, or at least in some of the abnormalities which are most severe. This increase in limb FA with severity could be either gradual/linear or could follow a threshold type of pattern, both of which will be tested in our analyses. Given the differences in the underlying developmental processes of the vertebrae and limbs, we study the link of DI in two different systems. If our hypothesis is confirmed, this can be seen as evidence that DI can be considered to be an organism-wide concept. If our hypothesis is not supported, this does not refute the whole paradigm that DI could reflect developmental stress and individual quality, but rather suggests that the link between general developmental processes and the asymmetry of particular traits is not general, but restricted to specific processes.

## MATERIAL & METHODS

### Subjects

Since 1980, all deceased infants and foetuses presented for autopsy at the VU University Medical Centre have been routinely radiographed both ventrally and laterally (23 mA, 70–90 kV, 4–12 s, Agfa (Mortsel, Belgium) Gevaert D7DW Structurix films). This research was carried out on the anterior-posterior projections of 1,389 deceased foetuses and infants obtained between 1990 and 2009. Not all babygrams were suitable for analysis, therefore we used the same selection criteria to include foetuses and infants in this research as described in detail in *Ten Broek et al. (2012)* and *Bots et al. (2014)*. In addition, some foetuses were excluded because of missing age information or our inability to measure limb FA (see
below). In total, we examined 528 male and 416 female foetuses and infants (13.7–92.1 weeks, mean: 27.9 weeks; standard deviation = 9.9 weeks). The babygrams were digitized using a Canon 30D digital camera in a fixed-distance set-up with a glass plate and a flash underneath.

## Vertebral variations

We examined the vertebral column of the foetuses for variation in both identity (i.e., cervical, thoracic, lumbar and sacral) and number of vertebrae. When ribs were present on the seventh vertebra, it was considered a transitional cervico-thoracic vertebra. Transverse processes of the seventh cervical vertebra exceeding those of the first thoracic vertebra (also known as apophysomegaly) were considered to be rudimentary cervical ribs fused with the transverse processes (e.g., *Pionnier & Depraz, 1956*; *Bots et al., 2011*). Ribs on the most caudal or most anterior thoracic vertebra were considered to be rudimentary and the vertebra to be transitional when the ribs were half the size of the preceding or subsequent rib, respectively (*Ten Broek et al., 2012*). We counted the number of vertebrae per vertebral region (i.e., cervical, thoracic, lumbar, and sacral) and classified transitional vertebra as having half the identity of both neighbouring regions, e.g., a transitional cervico-thoracic vertebra was scored as half cervical and half thoracic. Transitional vertebrae on the lumbo-sacral boundary were more difficult to score because in most individuals the sacral vertebrae were not yet fused with each other or with the ilium. However, shape and position of the vertebrae in the caudal region often provided adequate information, but still the presence of transitional lumbo-sacral vertebrae could have been underestimated.

Changes of the vertebral formula were expressed on a severity scale from 0–9 that reflected our estimations of the seriousness of the vertebral anterio-posterior (A-P) patterning disturbances, based on both the A-P position of the changes (i.e., the boundary) and the extent of the changes along the A-P axis (i.e., the number of boundaries, see for a detailed description *Ten Broek et al., 2012*). We have scored a regular (R) vertebral column with 7 cervical, 12 thoracic and 5 lumbar vertebrae as 0; a change in the number of lumbar vertebrae and, hence of the number of 24 presacral vertebrae, without other changes (lumbosacral: LS) as 1; lumbar ribs and absent or rudimentary twelfth ribs (thoracolumbar: TL) as 3; lumbar ribs and absent or rudimentary twelfth ribs with a changed number of presacral vertebrae (TL_LS) as 4; a cervical rib or rudimentary or absent first rib (cervicothoracal: CT) as 6; a cervical rib or rudimentary or absent first rib with a changed number of presacral vertebrae (CT_LS) as 7; a cervical rib or rudimentary or absent first rib with an absent or rudimentary twelfth rib, or lumbar rib (CT_TL) as 8; and a cervical rib or rudimentary or absent first rib with an absent or rudimentary twelfth rib, or lumbar rib and with a changed number of presacral vertebrae (CT_TL_LS) as 9 (*Ten Broek et al., 2012*).

## Asymmetry measurements and measurement error

We measured the length of the left and right femur, fibula, radius, ulna and tibia from the midpoint of the proximal end of the bone to the midpoint of the distal end of the bone. We also measured the left and right second and fourth digits by taking the length from the

**Table 1 Overview of levels of measurement error.** Measurement errors are the result of positioning the foetus, making the radiograph (ME-Radio), digitizing the radiograph (ME-Digitizing) and measuring a single radiograph (ME-measurement), and are relative to FA and levels of directional asymmetry (DA) in the different limb bones of the foetuses. The variance components were multiplied by 1,000.

| Trait | FA | ME-Radio | ME-Digit. | ME-Meas. | test for DA | DA p-value |
|-------|------|----------|-----------|----------|-------------|------------|
| Digit 2 | 0.60 | 0.35 | 0.00 | 0.00 | $X^2 = 0.93$ | 0.3361 |
| Digit 4 | 1.52 | 1.41 | 0.00 | 0.03 | $X^2 = 0.74$ | 0.3885 |
| Femur | 2.18 | 0.80 | 0.10 | 0.19 | $X^2 = 39.0$ | <0.001 |
| Fibula | 2.75 | 1.16 | 0.05 | 0.10 | $X^2 = 2.96$ | 0.09 |
| Radius | 1.17 | 0.51 | 0.29 | 0.14 | $X^2 = 2.97$ | 0.09 |
| Tibia | 4.95 | 0.36 | 0.08 | 0.08 | $X^2 = 0.00$ | 0.99 |
| Ulna | 1.43 | 0.71 | 0.00 | 0.24 | $X^2 = 4.03$ | 0.04 |

proximal end of the proximal phalanx to the distal end of the distal phalanx. We excluded all foetuses that had abnormally developed limbs from analyses, because otherwise the possibly higher measured FA could have been an artefact of the abnormalities directly. Four different investigators carried out all measurements without prior knowledge of the autopsy reports. Measurements were made in Image J version 1.42q. The images were spatially calibrated using a ruler that was present in the babygram. Thirty-one foetuses were re-measured independently by all examiners to ensure the accuracy of the measurements. Spearman's correlation tests showed that the left–right differences were highly comparable between the examiners (all $r > 0.30$ and all $P < 0.001$). In addition, the entire procedure of positioning and making the babygram was repeated for 147 individuals. A second independent digital photograph was made for 49 individuals and for 30 individuals the digital image was measured twice. These extra procedural steps allowed us to determine measurement error (ME) and directional asymmetry (DA) with a mixed model regression analysis using our measured bone lengths as response variable, side as a continuous covariate and both individual and the side-by-individual interaction as random effects (*Van Dongen, 1999*). Measurement error was smaller for all traits than the levels of fluctuating asymmetry and we found no directional asymmetry for all studied traits using $F$-tests, except for the femur and ulna (see Table 1 for values of measurement error, DA and FA). We obtained individual and trait specific asymmetry values after correction for DA and ME using the same mixed model regression analyses, as the best linear unbiased predictors (BLUPS) of the random slopes (*Van Dongen, 1999*). The unsigned asymmetry (i.e., the absolute value of these signed asymmetry values from the mixed regression models) correlated significantly with trait size for all traits (all $r > 0.20$ and all $P < 0.001$). Therefore we corrected FA measurements by dividing them by trait size, expressing trait-specific FA as a percentage. In a final step, we calculated the average asymmetry of the limbs for each individual. To make sure that each trait contributed an equal amount of information to this average FA, we first standardised each trait-specific asymmetry and then calculated the average across traits. In all analyses below we used this mean standardised FA as a measure of individual developmental instability (*Ten Broek et al., 2013* and *Bots et al., 2014*).

## Statistical analyses

We used linear models to explore the relationships between variations of the vertebral column and mean standardised FA, where standardised FA was used as dependent variable, and severity of variation of the vertebral column as explanatory variable. We added age (log-transformed) to the model as FA correlates negatively with age (*Van Dongen et al., 2009*; *Ten Broek et al., 2013*). We controlled for possible effects on FA in cases with deficient amniotic fluid volume by adding the presence or absence of sufficient amniotic fluid volume as a factor in the model (*Ten Broek et al., 2013*). The severity of variation in the vertebral column was tested as the continuous independent variable (assuming linearity) in one model and in a separate model as a factor (to test for non-linear effects). Average mean standardised FA was plotted for each of the eight categories. In addition, because foetuses with a regular vertebral column can have other congenital abnormalities, we also provide the mean standardised FA of foetuses without any congenital abnormality as a second reference group (but did not use this in any of the linear models). All analyses were performed in R version 3.0.2 (*R Core Team, 2013*).

## RESULTS

No linear increase of FA with severity of variation in the vertebral column was detected (slope: $-0.007 \pm 0.007$, $F_{1,939} = 1.38$, $P = 0.24$), after correction for age ($F_{1,939} = 2.1$, $P = 0.14$), amniotic fluid volume ($F_{1,939} = 6.9$, $P < 0.01$), and the interaction between age and amniotic fluid volume ($F_{1,939} = 7.3$, $P < 0.01$). In addition, limb FA did not differ between the eight indices of severity of variation in the vertebral column ($F_{7,933} = 1.03$, $P = 0.41$). Furthermore, exploring the average asymmetries, foetuses with a regular vertebral column had relatively high FA compared to foetuses with cervical ribs (CT) and additional abnormalities at other boundaries (TL and / or LS) though not statistically significant (Fig. 1). This also suggests that it is unlikely that we have missed a significant higher FA especially in these groups by chance. On average, FA was higher in the group of foetuses with a regular vertebral column even when considering only foetuses without developmental abnormalities (Fig. 1).

## DISCUSSION

We tested for an association between two markers of developmental instability (DI) in human foetuses, by studying associations between phenodeviance of the vertebral column and limb fluctuating asymmetry (FA). Unlike in other studies, we were able to score the severity of the abnormalities of the vertebral column on the basis of their associations with other congenital abnormalities (see *Ten Broek et al., 2012* for details). Therefore, if both phenodeviance and FA would measure individual DI, we not only predicted higher limb FA in foetuses with abnormalities in the vertebral column, but also an increase in FA with severity of these abnormalities. However, no such differences or association were detected, indicating that our two measures of DI—albeit both being related to other congenital abnormalities—reflect different aspects of developmental perturbations. We argue that both the differential timing of limb development and the patterning of the vertebral

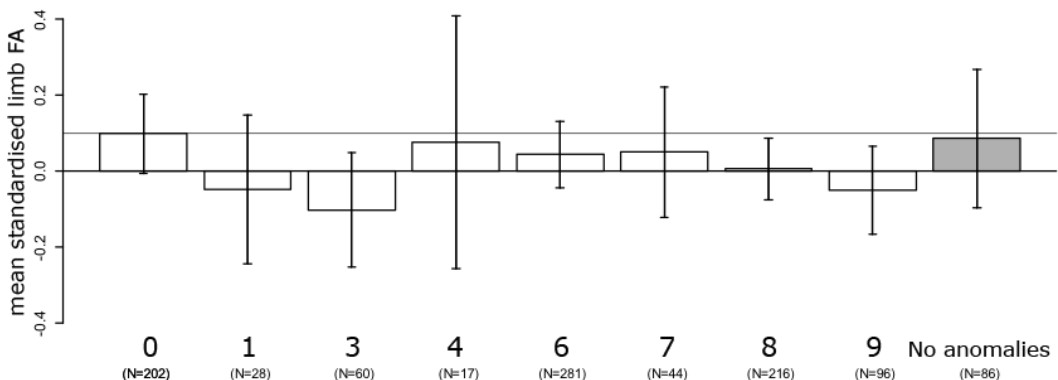

**Figure 1** **Average fluctuating asymmetry (mean standardised limb FA) in human foetuses and infants for the different groups of vertebral variation with increasing indices on the severity scale.** 0, Regular vertebral pattern; 1, Changes in the lumbosacral region; 3, Changes in the thoracolumbar and lumbosacral region; 6, Changes in the cervicothoracic region; 7, Changes in the cervicothoracic and lumbosacral region; 8, Changes in the cervicothoracic and thoracolumbar region; 9, Changes on all three boundaries of the vertebral column. For comparison, the horizontal line represents the control group (Regular). In addition, the last (grey) bar represents the average for all foetuses without vertebral abnormalities and no other major abnormalities. There were no significant differences among the different groups (see text).

column possibly are involved. Growth and differentiation during later development may well be more relevant for FA than the early patterning processes. Continued growth during the pregnancy and accumulation of effects of different sequential perturbations may blur relationships with specific disturbed developmental events earlier in life, in spite of the fact that the early perturbations have long lasting effects on other developmental processes like the vertebral variations do. Furthermore, the amount of buffering and possibilities for stabilising mechanisms to act may differ considerably. With respect to the vertebral axis, the identity of each element remains fixed after the anterior-posterior patterning of the paraxial mesoderm, while the continuous growth of limbs would allow to stabilise early asymmetric development at later stages. In sum, the development of the vertebral column and that of the limbs differ in so many aspects, with respect to timing, tissues and genes involved, that if developmental perturbations occur in either one of them, they do not seem to be interconnected genetically or developmentally. Limb FA and vertebral variations in human foetuses thus likely reflect developmental disturbances and are both positively correlated with the presence of some congenital abnormalities, yet do not signal individual wide DI. Our results thus contribute to the growing knowledge based on both empirical and theoretical work (for example recently reviewed in *Klingenberg, 2015*) that developmental perturbations in different developmental systems are not generally connected, or at least not in this case.

## ACKNOWLEDGEMENTS

We are grateful to Jaap van Veldhuisen and Ron Otsen of the photography division of the Department of Pathology of the VU University Medical Centre (VUMC Amsterdam) for the high-quality radiographs.

### Funding
The authors received no funding for this work.

### Competing Interests
The authors declare there are no competing interests.

### Author Contributions
- Clara M.A. ten Broek conceived and designed the experiments, performed the experiments, analyzed the data, wrote the paper, prepared figures and/or tables, reviewed drafts of the paper.
- Jessica Bots performed the experiments, reviewed drafts of the paper.
- Marianna Bugiani reviewed drafts of the paper.
- Frietson Galis conceived and designed the experiments, performed the experiments, reviewed drafts of the paper.
- Stefan Van Dongen conceived and designed the experiments, analyzed the data, wrote the paper, prepared figures and/or tables, reviewed drafts of the paper.

### Human Ethics
The following information was supplied relating to ethical approvals (i.e., approving body and any reference numbers):

Patient data and radiographs were used according the guidelines of the Medical Ethics Committee of the VU University Medical Center and patient anonymity was strictly maintained. Parental written informed consent was obtained for patients and data were handled in a coded and completely anonymous fashion, according to Dutch national ethical guidelines (Code for Proper Secondary Use of Human Data, Dutch Federation of Medical Scientific Societies; http://www.federa.org/codes-conduct).

### Data Availability
The raw data has been supplied as Supplementary Files.

### Supplemental Information
Supplemental information for this article can be found online at http://dx.doi.org/10.7717/peerj.3359#supplemental-information.

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
