# Peer review of "No relationship between vertebral column shifts and limb fluctuating asymmetry in human foetuses"

_PeerJ, doi:10.7717/peerj.3359_

## Round 0.1 · original submission · Minor Revisions

Dear Dr. Van Dongen,

I apologize for the delay getting back to you. It took me three rounds of requests to find suitable reviewers that would commit to a review over the holiday season. Both reviewers found this short communication to have promise and be potentially suitable for PeerJ. However, they request greater background on the nature of fluctuating asymmetry, the expectations and significance of the hypotheses, and discussion of the potential developmental causes for the lack of correlation between the two systems beyond a superficial statement on the differences in timing and gene expression.

Both reviewers also express concern about the availability of the raw data. The data set may be more useful to the readers if FA values for the individual anatomical sites were also included.

Reviewer 1 also requests further clarification of the standardization procedure in the calculation of the FA statistic (availability of the raw FA measurements may aid in understanding your process).

I have also provided an annotated pdf with suggested edits for grammar, typos and clarity.

In sum, these changes would seem to amount to a “minor revision.” If you thing you can address these concerns, I look forward to receiving a revised manuscript.

-Phil

·

Basic reporting

Generally, the provided references provide decent background for readers who are clinicians and biological scientists. However, it would be helpful to include some information on the possible mechanisms that are thought to underlie the normal differentiation of vertebrae or limb bones, as well as current hypotheses about the basis for the measured homeotic transformations in vertebrae or fluctuating asymmetry in limb length. While some of this information is provided in the discussion, fleshing out the introduction will provide a context for the hypothesis that these two measures of DI will be correlated. Along the same lines, it would also be helpful to more fully establish why understanding DI is important. If the hypothesis is supported, what does this mean more broadly? If is it not supported, what would this mean more broadly? Basically, the authors should further outline the basis for their hypothesis and why understanding whether two measures of DI are correlated is important.

There is a professional article structure. Raw data is shared. The results are definitely relevant to the hypothesis.

Professional English is used throughout, although there are a few errors in English within the Introduction. I make some suggestions that may clarify what the authors mean to say.

Line 17 – “between both” to “between these measures”
38 – “relates to environmental or genetic stress either” to “relate to either environmental or genetic stress”
40 – Period after DI
41 – indicator should be plural?
42-43 – change to “… cases. More specifically, the …”?
44 – does “more” mean “further”?
47 – change to “developing mammals generally” ?

Experimental design

This manuscript clearly fits within the Aims and Scope of the journal. The research question is well defined, although more might be done to illustrate why answering this question is important to understanding the developmental instability and/or the etiological basis of the traits under study. Please see my comments on the Introduction in Basic Reporting (above) for my suggestions about this.

The authors were careful to account for sources of measurement error, which is incredibly important when measuring fluctuating asymmetry. They also make valiant attempts to account for other known sources of variation in measured FA, including overall size of an individual. However, there are a couple of points that require more clarification.

First, starting on line 120, the authors indicate a final standardizing step while calculating average FA for each individual. It is not clear what mathematical operations are being applied here or what the data is being standardized for. This should be clarified. Because the input for this average is the FA of individual bones as a percentage of total bone length, I do not understand what other standardization step would be required to produce an average % FA of all bones.

Second, while the authors state that measurement error is less than the measure of FA, it would be valuable to have raw values of measurement error and FA reported. Although interobserver measures of left-right differences are significantly correlated, it looks like the correlation is not particularly strong (r = 0.30). What proportion of bone length does this type of error and other types of error account for? Similarly, what proportion of bone length do left-right differences account for in terms of both DA and FA? Providing this information will be a very useful context for any readers who are not familiar with asymmetry analyses.

Validity of the findings

Assuming that the final data standardization step makes sense, the conclusion is well stated and linked to original question. The speculation about the reasons for a lack of statistical relationship between vertebral features and limb FA are valuable and good food for thought.

As in the introduction, it would be useful for the authors to more explicitly state what the rejection of their hypothesis means for interpreting these measures of DI and for the conceptualization of proxy measures of DI in general. If each is not a measure of the same thing, then what should researchers in biomedical science, developmental biology, ecology, and paleontology take away from this?

Reviewer 2 ·

Basic reporting

The manuscript is well written and begins with a potentially interesting biological question. The authors report a negative result, but one that remains on biological significance. By far the most exciting elements of this manuscript is the data set on human embryology. However, only summary metrics are presented meaning that future authors will not have the ability to replicate this study or explore it further. I cannot recommend publication of this manuscript without public access to the raw data.

The authors do overlook a vast literature on fluctuating asymmetry that I strongly believe should be incorporated into the manuscript, particularly the work by Chris Klingenberg who has establish a great deal of the theoretical basis for these studies.

Experimental design

The authors take advantage of a unique series of human embryos that is unparalleled elsewhere in the world. I have no issues with the statistical analyses performed in this manuscript. However, and I reemphasize, I do feel that it is important for the raw data to be made available. Summary metrics are not appropriate.

Validity of the findings

The authors present a negative result - no correlation between limb asymmetry and vertebral defects - which appears robust. I do think that the authors should go into greater detail about the potential developmental bases of this. Concluding that this is simply because of differences in timing overlook the multitude of other ways that limb and vertebral column development is unique. In fact, there are far more differences than there are similarities at the gene and tissue levels of organization. I, for one, am not surprised by this lack of correlation.

---

## Round 0.2 · Minor Revisions

Thank you for resubmitting your revised manuscript. I sent this for re-review to one of the previous reviewers. In many respects it is improved. However, the greater clarity of the revised manuscript has raised further issues regarding the methodology. These need to be addressed before final acceptance. As you can see, the reviewer sees that there is the potential that significant changes will be needed to address these issues. I apologize that these were not raised in the previous review, but the brevity of the previous submission prevented full assessment of your statistical analysis. I look forward to your resubmission once these issues are addressed.

·

Basic reporting

The additions and modifications to Introduction, Methods, and Discussion add further clarity and an explicit hypothesis to a manuscript that was already professional with a clear structure.

Experimental design

The research questions are well defined and interesting. Seeking to understand whether developmental instability has an organism-wide systemic effect is important for understanding how developmental perturbations (either genetic or environmental) might influence overall morphology.

The authors make a good effort to quantify the measurement and imaging error, which is important. They have clarified these and some of their other methods. With this deeper understanding, I note several potential issues with their measurement of FA. Measurement of vertebral phenotype seem quite appropriate.

If I understand the methods correctly, FA is measured for each individual and then for each group in the following way. It is possible that I have misinterpreted the method used. If so, I apologize.
1) A mixed-model regression analysis is used to determine the effect of directional asymmetry (DA) and measurement error (ME) on phenotype (although this model or any related results are not reported at all).
2) Limb measurement errors are run through the model and the regression residuals are used for the rest of the analysis under the idea that the effects of DA and ME have been corrected for. I assume that the measures input are raw linear distances, but this is not made clear in the methods.
3) For all bones/specimens for which left and right measurements exist, the residual of left side length is subtracted from the residual of right side length (or vice versa). The result is an asymmetry measure for each bone for each individual. These values can be positive or negative.
4) These asymmetry measures are standardized by average trait size (I assume average residual trait size, but it is not clear). Then, they are averaged for each individual, which means that negative and positive values are averaged together. Finally, these individual averages are averaged by group. Once again, a mean of negative and positive values (as evidenced by pos and neg means in Figure 1).

If I understand correctly, two major issues are raised...

A) We do not expect that these mean measures of Left-Right divergence, which arguably can be interpreted as individual measures of FA, to be different then zero. In fact, if there is no directional asymmetry, regardless of the presence of FA in the population, we would expect them always to be zero. Therefore, we would never expect any of our groups to differ by this measure. Fluctuating asymmetry is by definition random, which means that in populations with strong or weak FA, the mean left-right difference will be zero (see Klingenberg 2015, Figure 4A). Therefore, if you wish to compare this measure of divergence from symmetry, you should use an absolute value of left-right difference. If you do this, you should only have positive means for your groups (all positive in Figure 1). I am not certain this measure should be called FA, but I understand why you are interpreting it as such. Fluctuating Asymmetry is typically a population based measure.

B) Is there significant FA within this population or within any of the groups specifically? If not, the individual asymmetry comparisons might just be comparisons of noise. However, this could be interpreted similarly to how you the current negative result is interpreted. A test for significance of FA is strongly suggested. In order to do this, simply take use a mixed model regression (or MANOVA) with the following factors: individual, side (DA), individual x side interaction (FA), measurement error. So, this should just be expanding the current regression model. If you do this, you should be able to report on whether DA or FA are significant factors in your population. You might also run this seperately for the various vertebral groups or perhaps use some sort of vertebral group interaction term instead. This might help determine if there are differences in FA between your groups. This might be a more common measure of FA than what is described in A).

In addition to the suggestions in A) and B):
- make sure to include some further details to clarify confusion regarding the interpretation of methods (as mentioned in 1-4, above)
- include more details of the significance and effect size of terms in any regression model you test, including the DA/ME one currently described and/or the one suggested in B). This information is critical to interpretation of the presence and influence of DA or FA in this population.

Validity of the findings

The results appear valid, but may be modified, depending on methods changes.

Additional comments

I have some additional suggestions for wording that may help clarify what the authors mean.

line 17 & 20 - genome-wide is probably not what the authors mean. I prefer the use of organism-wide as is found at the end of the introduction. As far as I can tell, there is no idea that the whole genome is modified. Instead, the question is whether DI influences all parts of the organism similarly, given a genetic or environmental perturbation.
36 - "of on average" change to "between"
51-53 - Is the following better? "More specifically, changes in vertebral identity along the vertebral column provide a good ..."
62 - is the following better? "...while a vertebral abnormality in of itself poses no ..."

---

## Round 0.3 · Minor Revisions

Thank you very much for throughly responding to the reviewers' concerns. The paper is essentially accepted. I have made a few grammatical and typographical edits in a tracked Word file (saved as a pdf). Once the manuscript is accepted it goes straight to production. Read through it one last time, including the references, and it will be good to go. Thank you very much for submitting this very interesting work!

---

## Round 0.4 · accepted · Accept

Thanks for the quick return of your manuscript and sorry for the confusion regarding annotated comments.